# Study of the Explosive Bridge Film Using Laser Shaping Technology

**DOI:** 10.3390/mi13060854

**Published:** 2022-05-29

**Authors:** Dangjuan Li, Siyu Li, Kexuan Wang, Junxia Cheng, Jia Wang, Shenjiang Wu, Junhong Su

**Affiliations:** 1School of Photo-Electrical Engineering, Xi’an Technological University, Xi’an 710021, China; licy@xatu.edu.cn (D.L.); m13772633470@163.com (S.L.); chengjunxia111@163.com (J.C.); wangjiar1001@126.com (J.W.); 2Shannxi Applied Physics and Chemistry Research Institute, Xi’an 710061, China; sxwkx@126.com

**Keywords:** shaping, laser damage, laser repair, explosive bridge film

## Abstract

Laser shaping technology and its applications have gained widespread attention in different fields. Using laser repair technology prolongs the service life of micro-explosive products and reduces the production cost, as well as enables the recycling of resources. Although most research mainly focuses on aspheric surface shaping and testing technology, only a few studies on repair technology for micro-explosive products using laser shaping have been reported. To promote the better application of laser shaping technology in the production and repair process of micro-explosive components, this work mainly studied the effect of laser shaping on the repair of an explosive bridge film to enhance the ignition performance and prevent damage. Different processes were used to repair the metal film using laser shaping and non-shaping, respectively. Furthermore, we investigated the similarities and differences of a laser-damaged film surface before and after shaping, and the influence of laser energy parameters on the microstructure and ignition properties of the repaired region. Additionally, we obtained a reasonable repair scheme by analyzing the temperature field variation from the simulation. The results show that the damage caused by the non-shaping and shaping lasers can be repaired using the heat flow and vaporization methods, respectively. By controlling the process parameters, the quality of repair can be improved and the production cost of the bridge film can be reduced.

## 1. Introduction

When the micro-ignition device is energized, the heat of the energy converter causes the thin film to generate transferrable energy to the micro-initiation charge through the aligned gap, forming the initial ignition. Meanwhile, the metal bridge film is prone to wear and forms defects during processing and preparation. This damage affects the operation of the entire ignition system and the safety and reliability of the equipment. Moreover, the high repair value is attributed to the high price of explosive components, complex preparation process, and long production cycle. Therefore, repairing micro-explosive components has become an important technical problem requiring an urgent solution. Since the femtosecond laser can process the surface of a material and the interior of a transparent material [1], using laser repair technology not only prolongs the service life of micro-explosive products and reduces production costs but also recycles resources.

The laser damage experiment showed that the damage at the center of Gaussian laser damage morphology was serious. To evenly damage and repair them, it is necessary to shape the laser spot [2]. Many researchers have researched the application of laser shaping technology in different fields. Pan Yisi et al. (2010) [3] obtained the vector height function of the front and rear surfaces of an aspherical lens using the nonlinear minimum quadratic function and Romberg numerical integration method, which was applied to a CO_2_ laser. The optical shaping system overcomes the shortcomings of the traditional two-piece shaping system and has the characteristics of a small volume and an easy adjustment. Yin Zhiyong et al. designed a semiconductor laser shaping system using a microlens array of hyperboloid substrate [4]. Li Rui et al. (2015) [5] studied the design method for the aspheric shaping system, which obtained the flat-top light with a diameter of 60 μm using the shaping laser. They made the flat-top light more effective in controlling the thermal effect using a femtosecond laser micro-machining experiment. M. Slimani et al. (2013) [6] shaped a Gaussian beam with a wavelength of 808 nm and a power of 1.2 KWs into a flat-topped beam, and finally obtained a linear spot. They used ZEMAX software to simulate the shaping process, which indicated that the size of the linear spot was about 10 mm × 0.5 mm. Moreover, the energy uniformity of the spot along the slow axis direction was up to 95%. Shi Guangyuan et al. (2014) [7] proposed an axis-symmetric Gaussian beam shaping system composed of two aspherical cylindrical mirrors placed in the orthogonal direction. The intensity uniformity of the rectangular beam after shaping was higher than 95%, with an approximate 10% energy loss. Liu Dan (2015) [8] elaborately designed two schemes by shaping Gaussian beams into linear spots. Here, both the Powell prism and Fresnel lens were used to obtain the linear spot, which meets the requirements of use and has been successfully applied in laser ultrasonic excitation and detective technology. Lindle et al. studied an efficient multibeam large-angle nonmechanical laser beam steering from computer-generated holograms rendered on a liquid crystal spatial light modulator [9].

The above research mainly focused on aspheric surface shaping and testing technology. However, only a few studies on laser repair technology for micro-explosive products using laser shaping have been reported. To promote the better application of laser shaping technology in the production and repair of micro-explosive components, this paper takes copper plated on silicon material as the research object. From the theory of laser shaping technology, we used a Powell prism and ZEMAX software for the simulation. Furthermore, the surface microscopic characteristics and ignition performance of the film before and after repair were analyzed by combining experiments with simulation. These findings will provide a theoretical basis and technical support for improving the quality of the laser repair of micro-explosive film surfaces.

## 2. Theory and Method

### 2.1. The Materials and Methods

Cu explosive bridge film was deposited with a designed mask on double-sided polished silicon (P-type, 100-grain direction) using a magnetron sputtering machine (MSP-400 B). The target diameter of 60 inches and purity of 99.99%, were fixed on the sidewall of the chamber. The sample table was rotated at 30 rad/min and kept water-cooled. Before the film deposition, all the substrates were placed in a 3:1 mixture of alcohol and diethyl, cleaned for approximately 30 min using an ultrasonic technique, and then, air-dried in a vacuum chamber. The cleaning system used a working vacuum of 2 × 10^−1^ Pa and an ion beam of 50 mA with an energy of approximately 1500 eV. After 30 min of cleaning, the ion-cleaning source was switched off, and the argon gas supply was stopped for the pulsed arc to deposit the films. A sputtering power of 200 W and an argon gas flow of 30 sccm were used for Cu film deposition. The deposited Cu explosive bridge film with holes and scratches is shown in Figure 1. The laser damage threshold test system was used to damage and repair the film before and after shaping, as shown in Figure 2.

To obtain the shaping pulse waveform, a Powell prism was designed using ZEMAX. Then, the designed prism was used to shape the Gaussian laser into a flap-top style. The Powell prism is an aspherical cylindrical mirror that can effectively eliminate the edge distribution and central hot spot effects of Gaussian beams [10]. The laser can be shaped into a straight line by a single Powell prism or a rectangular spot by a composite system of Powell beams. The latter is preferred when the local defect point of the Cu bridge film needs repair. However, when the entire Cu bridge film is being repaired, laser spot shaping as a straight line cooperating with a straight line slide platform has higher work efficiency.

The sample morphology was tested using a profilometer and microscope. Damage detection adopted the He–Ne scattering method to ascertain the damage degree with respect to changes in scattering energy [11]. The principle of laser irradiation repair is that the material absorbs part of the energy when it is irradiated by a laser. With the different power densities of the irradiated laser comes a difference in the temperature of the material, thus a series of physical changes occur. In the beginning, the irradiation laser energy is lower because heating just started. When the irradiation energy increases, melting flow occurs on the surface of the material. Furthermore, a continuous and gradual increase in energy propels vaporization to occur on the surface of the material. This leads to a rise in the energy of the irradiated laser, after which the phenomenon of plasma sputtering appears. Laser repair mainly uses the melting flow and evaporation processes. Melting repair is the use of laser irradiation on the surface of the optical element defect or damage point. Therefore, when the material in the damaged area reaches the melting temperature, melting flow occurs, and the damaged point is repaired to a certain extent. Evaporative repair uses a laser to irradiate defects or damage points on the surface of optical elements to make the damaged area reach the gasification temperature. This removes part of the damaged area to achieve repair [12]. It deposits several materials on the surface of components after the vaporization of materials in the damaged area, whereas non-evaporative repair (melting repair) will not produce material deposition by considering the characteristics of material melting flow [13].

### 2.2. Basic Theory

From the theoretical calculation of the energy required by the experiment, the temperature (*T*) of the center (*r* = 0) of the surface (*z* = 0) of the damaged area is [10]:(1)$$T(0,0,t,J)=\frac{AJr_{0}^{2}}{\rho c\sqrt{\pi \alpha_{d}t}\left(4\alpha_{d}t+r_{0}^{2}\right)}$$

Here, *r*, *z*, *k*, *ρ*, *c*, and *A* are the radial coordinate, damage depth, thermal conductivity, density, specific heat, laser irradiation area, respectively.

When the laser pulse width approaches zero, i.e., *t_p_*→0, the diffusion coefficient: $\alpha_{d}=\frac{k}{\rho c}$. Thus, substituting into Equation (1), we obtain:(2)$$T(0,0,t,J)=\frac{\sqrt{\alpha_{d}}AJ}{k\sqrt{\pi t_{p}}}$$

When the temperature rises to the melting temperature *T**_m_*, the minimum energy (*J*_min_) required is
(3)$$J_{\min}=\frac{T_{m}k\sqrt{\pi t_{p}}}{\sqrt{\alpha_{d}}A}$$

When the temperature rises to the gasification temperature *T_g_*, the minimum energy required is
(4)$$J_{\min}\frac{T_{g}k\sqrt{\pi t_{p}}}{\sqrt{\alpha_{d}}A}$$

## 3. Results Analysis and Discussion

### 3.1. Shaping Design

The ZEMAX software was used for the simulation to obtain the flat-top beam. The Powell prism was selected for laser shaping, and the diameter of the flat-topped beam was slightly smaller than that of the Powell prism [14]. In ZEMAX, the entry pupil diameter of the optical system was set as 4 mm, and the type was Gaussian. The initial incident angle equals zero, i.e., both the x and y directions in the field of view were zero. The sector size of the light emitted from the Powell prism was related to the refractive index of the glass and the angle of the prism roof. Meanwhile, the sector size affects the length of the optical stripe at the same projection distance, which is inversely proportional to the angle of the prism surface and proportional to the refractive index of the glass. We used the BK7 glass with a refractive index of 1.5168, which met the requirement of the experiment. Table 1 shows the parameters of the Powell prism after simulation using ZEMAX, and Figure 3a,b show the light power distribution before and after shaping. Thus, we obtained a Powell prism that can shape the Gaussian beam.

### 3.2. Damage Micro-Topography under Different Laser Energy

The copper film was fixed on the loading platform of the laser damage threshold system, in which the laser parameters were controlled by a computer to test the threshold value. Different laser energy damages occurred by changing the light transmittance. Then, a profilometer was used to scan the copper-coated silicon damage using a non-shaping laser. Next, optical imaging was conducted using the interference principle to obtain the microstructure after damage. Figure 4a–f showed the micro-topographies of different laser energies at 181, 127, 90, 36, and 18 mJ, respectively. The surface roughness (Sa) was 339.2 nm when the damage energy was 181 mJ. Furthermore, as the energy decreased, Sa also gradually decreased. With the decrease in the laser damage energy comes a decrease in the damaged area and degree, respectively. When the energy is lower, the damage appears in the center more obvious, indicating that the energy is more concentrated. When the energy increased, the peak value of damaged morphology also increased, similar to the energy distribution of the Gaussian laser. Additionally, when the laser energy was 36 mJ, the maximum damage peak difference was about 8.6 um. From the laser repair threshold, molten flow occurs when the laser energy is greater than 41.45 mJ. When it is greater than 97.3 mJ, vaporization and evaporation will occur, which results in the partial repair of the central damaged area. Thus, the damage peak difference is relatively smaller.

### 3.3. Damaged Surface of Shaping and Non-Shaping Lasers

After the simulation of the Powell prism was made, it was loaded into the damage tester system, and then the laser was used to conduct the damage test. The damaged morphology of the non-shaping and shaping laser was compared, as shown in Figure 5 and Figure 6. It can be seen that under the same laser energy, the damage of the shaping laser is less than that of the non-shaping laser in the difference in damage peak values at 507.2 and 1212.2 nm, respectively. That is, its ratio is about 1:2.4. This indicates that the damage threshold of the shaping laser to the material improved. Additionally, the damaged surface of the shaping laser does not show a Gaussian distribution, which is similar to a strip shape. The damaged area caused by the unshaped laser is mainly concentrated on the ring and the damage energy is focused, but not concentrated, on the central point. This is because the light path is not centered due to the height of the convex lens, which affects the energy collection.

### 3.4. Simulation of the Temperature Field Distribution

COMSOL software was used to simulate ignition performance using the temperature field distribution. From the finite element method, we obtained the physical phenomena realized by solving partial differential equations of the multi-physics field. The finite difference element method was used to solve current and thermodynamic problems. The meshing of the model structure for copper plated on silicon is shown in Figure 7. Here, the thickness of the film is 2000 μm and the substrate is 2500 μm. Table 2 shows the laser and material parameters.

Figure 8 and Figure 9a–c show the simulation results. Observe that the damage depths of the non-shaping and shaping lasers were about 100 μm and 50 μm, respectively. That is, the ratio of the two damages equals 2:1, which is consistent with our experimental test results. Moreover, the surface of the non-shaping laser damage is convex, whereas that of the shaping laser damage is flat. This suggests that different laser shapes can be used to adjust and control the desired damage. When the same laser energy (60 W) was used, the damaged area of the shaping laser was smaller than that of the non-shaping laser. The damaged areas for 6 and 20 W conditions are very similar. Meanwhile, when the damage is maximum by different lasers, the time required and the temperature reached are different, i.e., the non-shaping laser reached 850 K for 7 μs, and the shaping laser reached 1100 K for 5 μs. Additionally, for the shaping laser, different energies can achieve different ignition temperatures, such as 378 K at 6 W, 555 K at 20 W, and 1100 K at 60 W, with the required time being 4.5, 4.1, and 5 μs, respectively. Therefore, the larger the laser energy, the higher the ignition temperature and the longer the ignition time. Therefore, the ignition performance of the bridge film using the repaired shaping laser outperforms that of the repaired non-shaping laser.

### 3.5. Repair of Damage

In the experiment, the laser repair technology is mainly aimed at small size damage points and laser irradiation. The damage point was repaired by irradiated melting to reduce the damage size. The Nd_YAG laser of the Gaussian pulse was applied with a wavelength of 1064 nm, a pulse width of 10 ns, and a maximum spot radius of 0.4 mm. When the energy is relatively low, the damage is concentrated at the spot center [15]. Furthermore, when the energy increases, the damaged area shows the spot shape. However, by adding the optical shaping system, the damage threshold increased under the same energy. The damage depth distribution is not high in the middle and decreased on both sides, but the overall distribution was more uniform. Therefore, at low energy, the non-shaping laser can be repaired using the heat flow method. Figure 10 shows the film surface morphology before and after the shaping laser repair. Combined with the simulation data, the laser energy of the thermal flow repair was 41.45 mJ, and the energy of evaporation was 97.30 mJ. The ignition performance was simulated according to the temperature field variation before and after repair. It was found that the shaping laser had little repercussion and an improved overall effect.

## 4. Conclusions

In this paper, we investigated the damage and repair of a copper-plated film using shaping and non-shaping lasers. First, we designed the Powell prism using ZEMAX and the Gaussian laser was shaped. Furthermore, we established the laser shaping optical system and performed a laser damage experiment. The laser damage and repair problems in the processing of the explosive bridge film were analyzed. The results show that the shaping laser is more convenient for repair and reduces the production cost. The specific conclusions are as follows:The effects of non-shaping and shaping laser damage energy on the surface morphology and ignition performance of copper films were analyzed by microscope and profilometer observation. The results showed that the surface roughness decreased with the decrease in laser energy, and the damaged area and damage depth increased with the increase in laser energy. Furthermore, the lower the energy, the more the damage was at the center than in the surrounding area. Moreover, the damage morphology was the same as the energy distribution of the Gaussian laser. The degree of damage under the shaping laser was relatively uniform.The simulation analysis showed that the temperature of the explosive bridge film increases with an increase in laser energy. Moreover, the shaping laser has a smaller damage area and a shorter laser action time than the non-shaping laser with equal energy. However, the ignition temperature of the explosive bridge film was 1100 K for the shaping laser and 850 K for the non-shaping laser.When the energy is relatively low, the non-shaping laser damage can be repaired using the heat flow method and the repair threshold of 41.45 mJ. For the shaping laser damage, the repair effect is better by the vaporization method with the repair threshold of 97.3 mJ.The shaping laser outperforms the non-shaping laser in repairing the defects or injuries on Cu explosive bridge film.

## Figures and Tables

**Figure 1 micromachines-13-00854-f001:**
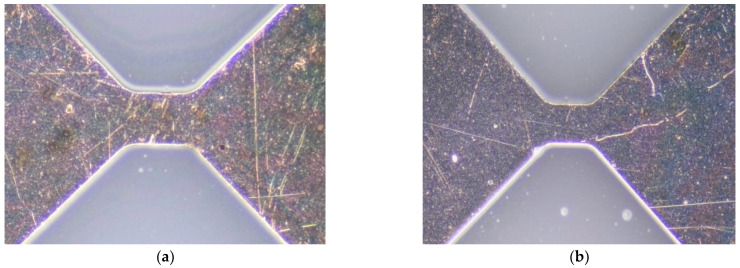
Cu explosive bridge film with hole (**a**) and scratch (**b**).

**Figure 2 micromachines-13-00854-f002:**
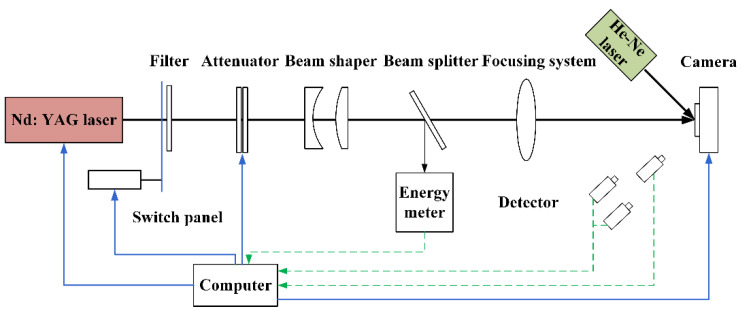
Laser damage threshold test system.

**Figure 3 micromachines-13-00854-f003:**
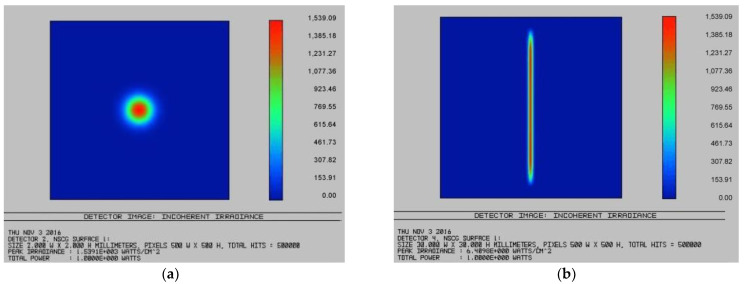
Laser power distribution before shaping (**a**) and after shaping (**b**).

**Figure 4 micromachines-13-00854-f004:**
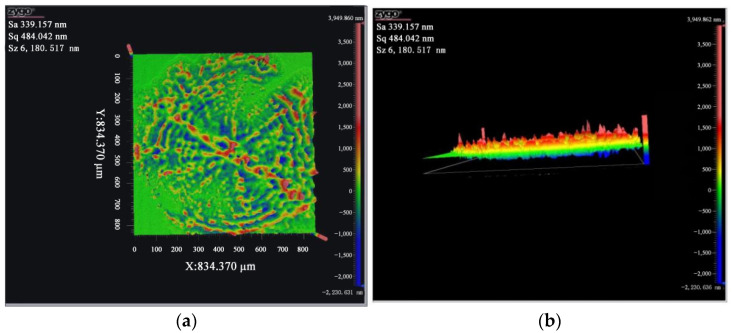
Micro-topography of different laser energy damage. (**a**) 181 mJ (top view), (**b**) 181 mJ (side view), (**c**) 127 mJ, (**d**) 90 mJ, (**e**) 36 mJ, and (**f**) 18 mJ.

**Figure 5 micromachines-13-00854-f005:**
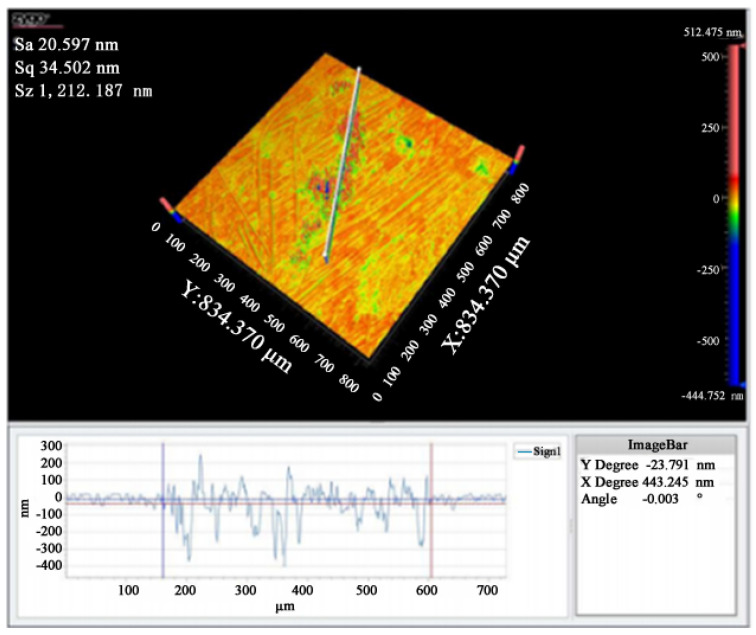
Damage surface and characterization under a non-shaping laser.

**Figure 6 micromachines-13-00854-f006:**
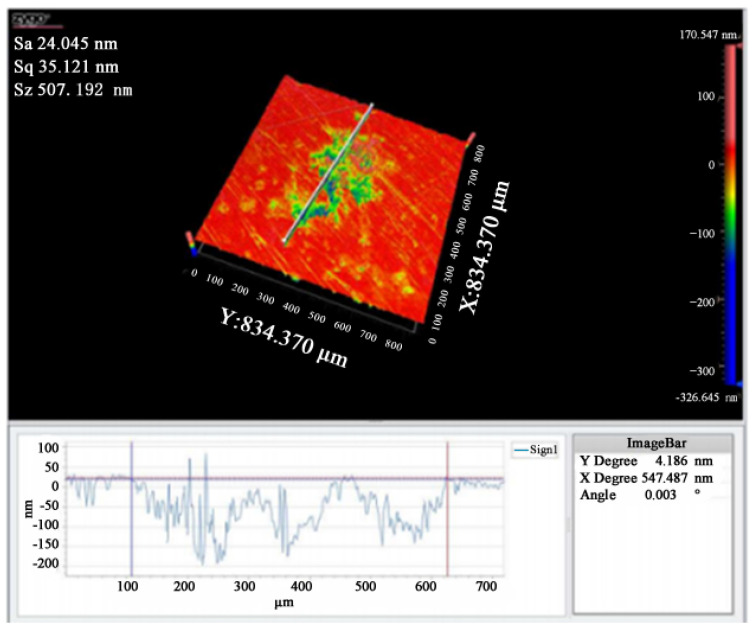
Damaged surface and characterization under shaping laser.

**Figure 7 micromachines-13-00854-f007:**
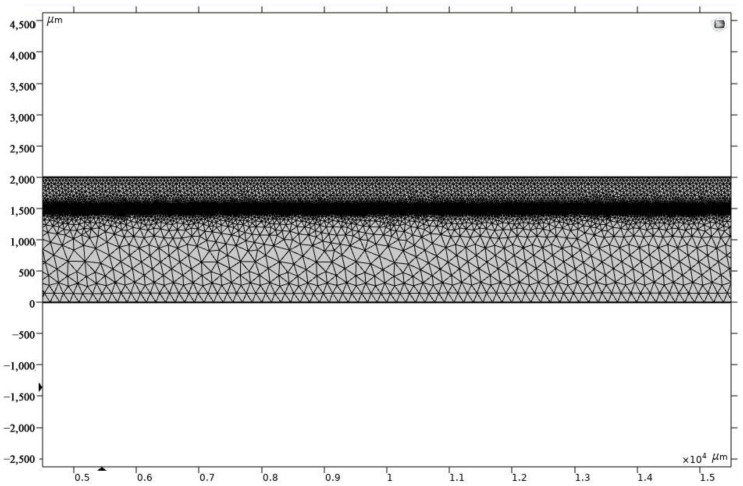
Meshing of the model structure of copper plated on a silicon wafer.

**Figure 8 micromachines-13-00854-f008:**
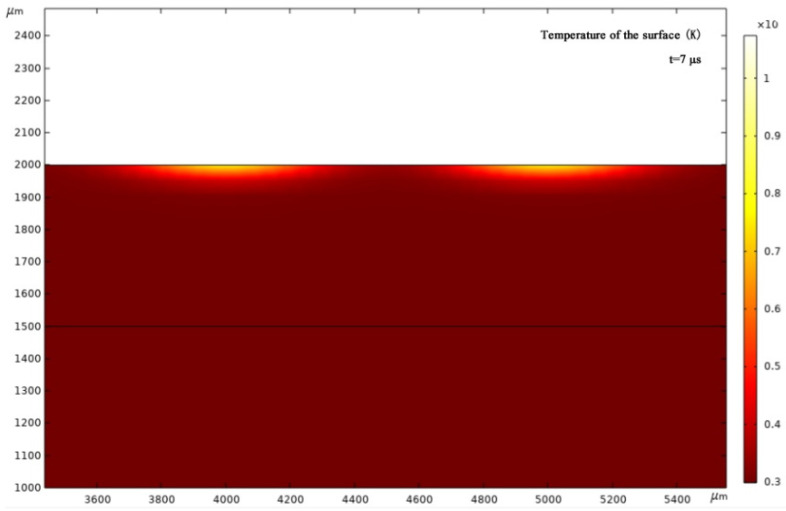
Temperature field distribution under non-shaped laser energy 60 W.

**Figure 9 micromachines-13-00854-f009:**
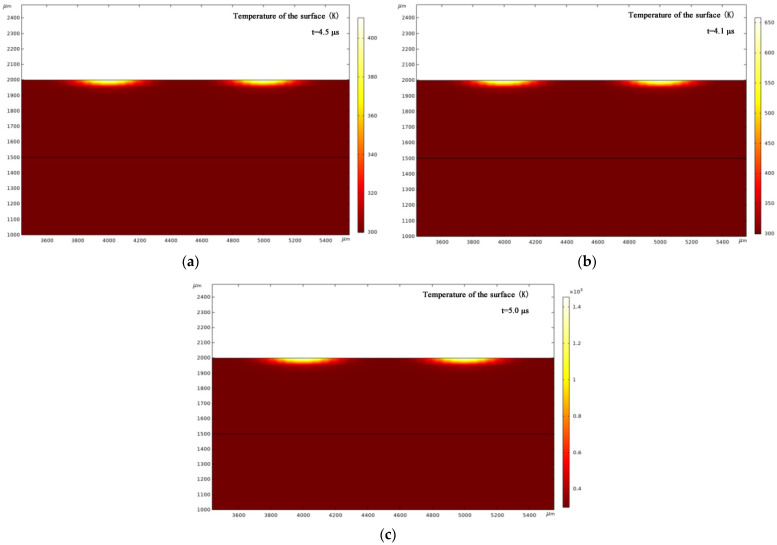
Temperature field distribution under different energies of the shaping laser. (**a**) 6 W; (**b**) 20 W; and (**c**) 60 W.

**Figure 10 micromachines-13-00854-f010:**
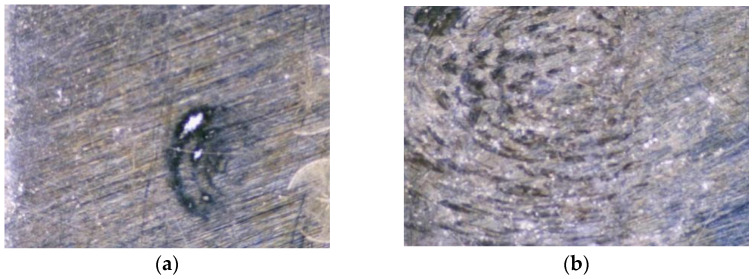
Film surface morphology under (**a**) before repair and (**b**) after repair with shaping laser.

**Table 1 micromachines-13-00854-t001:** The parameters of the Powell prism.

Type	Radius of Curvature	Thickness	Glass	Diameter	Conical Coefficient
Cylinder	2 mm	5 mm	BK7	3 mm	−1.600

**Table 2 micromachines-13-00854-t002:** The input parameters of material and laser.

Symbolic	Expression	Value	Name
H_s	205 (kJ/kg)	2.05 × 10^5^ J/kg	Sublimation heat
T_a	900 (degC)	1173.2 K	Burning temperature
rho	8900 (kg/m^3^)	8900 kg/m^3^	Density
Cp	390 (J/kg/K)	390 J (kg·K)	Specific heat
k	397 (W/m/K)	397 W/(m·K)	Coefficient of thermal conductivity
r_sport	20 (μm)	2 × 10^−5^	Spot radius
x0	4000(μm)	4 mm	Left spot position
x1	5000 (μm)	5 mm	Right spot position
P_total	1 (W)	1 W	Total laser power
P_density_avg	P_total/(pi*r_sport^2*H)	39,789 W/m^2^	Average laser power
H	20,000	20,000	Laser frequency

## Data Availability

The data that support the findings of this study are available from the corresponding author upon reasonable request.

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
