# Peer review of "Study of the Explosive Bridge Film Using Laser Shaping Technology"

_micromachines, 2022, doi:10.3390/mi13060854_

Round 1
Reviewer 1 Report
Title: Study on the explosive bridge film by laser shaping technology
ID#: 1705338
This work reports the numerical analysis and experimental studies on the damages of the explosive bridge film by shaping and non-shaping lasers and the repair technologies. The comments as below should be addressed before further consideration.
1. Paragraph from line #91 to line #110 represents the nature of two different repair techniques. The authors should indicate the advantages and disadvantages of each method, that may relate to the cost optimization.
2. For the subsection 2.2, the used parameters must be given with their meaning. Some of them appear in Table 2, but it is better to clarify the meaning of each parameter at its first mentioned in the manuscript.
3. In subsection 3.4, paragraph from line #190 to line #204, the authors applied three different shaping lasers. How about the results for non-shaping laser with the same energy?
4. The caption of Fig. 9 should clarify. Are the results of non-shaping or shaping lasers?
5. In section 3.5, only experiment of repairing by irradiated melting is considered, thus, it is not reasonable to state that “For shaping laser, the effect of vaporization is better”.
6. There are the typo errors; the English are weak; the authors should check and revise for enhancing the manuscript.
Author Response
Revision lists and replies to the reviewer’s comments
Re:
Manuscript ID#: : 1705338
Title: Study on the explosive bridge film by laser shaping technology
Authors: Dangjuan Li, Siyu Li, Kexuan Wang, Junxia Cheng, Jia Wang, Shenjiang Wu, and Junhong Su
We are grateful to the Editor and the two anonymous reviewers for their comments and suggestions. The corresponding revisions and replies to the manuscript have made and listed as follows.
The comments and replies:
Reviewer #1:
Comments and Suggestions for Authors
Title: Study on the explosive bridge film by laser shaping technology
ID#: 1705338
This work reports the numerical analysis and experimental studies on the damages of the explosive bridge film by shaping and non-shaping lasers and the repair technologies. The comments as below should be addressed before further consideration.
- Paragraph from line #91 to line #110 represents the nature of two different repair techniques. The authors should indicate the advantages and disadvantages of each method, that may relate to the cost optimization.
Reply: Relevant explanations were given in Paragraph 2.1 line #100 to line #103.
- For the subsection 2.2, the used parameters must be given with their meaning. Some of them appear in Table 2, but it is better to clarify the meaning of each parameter at its first mentioned in the manuscript.
Reply: We have correct the missing information.
- In subsection 3.4, paragraph from line #190 to line #204, the authors applied three different shaping lasers. How about the results for non-shaping laser with the same energy?
Reply: laser energy of 6W, 20W, 60W were used for simulating the damage area of shaping laser and non-shaping laser. When the same laser energy (60W) was used, the damage area of shaping laser is smaller than that of non-shaping laser. The areas of damage in the other 6W and 20W conditions are very similar and are therefore not given in the manuscript.
We have supplemented this in the paragraph.
- The caption of Fig. 9 should clarify. Are the results of non-shaping or shaping lasers?
Reply: We have correct the missing information.
- In section 3.5, only experiment of repairing by irradiated melting is considered, thus, it is not reasonable to state that “For shaping laser, the effect of vaporization is better”.
Reply: Thanks for reminder. We have delete the statement of “For shaping laser, the effect of vaporization is better”.
- There are the typo errors; the English are weak; the authors should check and revise for enhancing the manuscript.
Reply: Thank you very much for your helpful comments. We have invited a professional institutions (www.enago.com) help us revise this paper. The authors would like to thank Enago (www.enago.cn) for the English language review.
In addition, we also systematically corrected the full manuscript and redrew part of the figure.
Thank you very much.
Sincerely yours,
Shenjiang Wu

Reviewer 2 Report
The reviewed work is a research on focusing laser light over a thin metal film. In this relation, I find important the following points:
- The Authors need to substantiate the use of copper as the film material and to specify the thickness of the film.
- The Conclusion should explicitly formulate the Authors’ opinion as to positive or negative effect of introducing a thin metal film into the focal spot for various applications.
If these points are addressed in a further revision of the manuscript, it may be published in Micromashines.
Author Response
Revision lists and replies to the reviewer’s comments
Re:
Manuscript ID#: : 1705338
Title: Study on the explosive bridge film by laser shaping technology
Authors: Dangjuan Li, Siyu Li, Kexuan Wang, Junxia Cheng, Jia Wang, Shenjiang Wu, and Junhong Su
We are grateful to the Editor and the two anonymous reviewers for their comments and suggestions. The corresponding revisions and replies to the manuscript have made and listed as follows.
The comments and replies:
Reviewer #2:
Comments to the Author
- The Authors need to substantiate the use of copper as the film material and to specify the thickness of the film.
Reply: Thank you very much for your helpful comments. According to the suggestion, the acronyms have been rewritten in the revised version. Some words have been added to explain the acronyms. The deposition method, conditions, processing parameters, etc. have been added in Paragraph 2.1. The deposited Cu explosive bridge film were shown in Fig. 1. Laser damage threshold test system shown in Figure 2 was redrawn.
- The Conclusion should explicitly formulate the Authors’ opinion as to positive or negative effect of introducing a thin metal film into the focal spot for various applications.
Reply: Relevant explanations were given in Paragraph 2.1 line #100 to line #103.
Thank you very much.

Round 2
Reviewer 1 Report
The reviewer can see the improvement of the revised manuscript. It has an impact and adapts the requirement of the journal.
It would be accepted for publication in Micromachines.